# Chromosomal Instability Causes Sensitivity to Polyamines and One-Carbon Metabolism

**DOI:** 10.3390/metabo13050642

**Published:** 2023-05-09

**Authors:** Anowarul Islam, Zeeshan Shaukat, David L. Newman, Rashid Hussain, Michael G. Ricos, Leanne Dibbens, Stephen L. Gregory

**Affiliations:** 1College of Medicine and Public Health, Flinders University, Adelaide 5042, Australia; 2Clinical and Health Sciences, University of South Australia, Adelaide 5001, Australia; 3School of Biological Sciences, University of Adelaide, Adelaide 5006, Australia

**Keywords:** s-adenosyl methionine, spermine, genomic instability, reactive oxygen species (ROS), *Drosophila*, autophagy

## Abstract

Aneuploidy, or having a disrupted genome, is an aberration commonly found in tumours but rare in normal tissues. It gives rise to proteotoxic stress as well as a stereotypical oxidative shift, which makes these cells sensitive to internal and environmental stresses. Using *Drosophila* as a model, we investigated the changes in transcription in response to ongoing changes to ploidy (chromosomal instability, CIN). We noticed changes in genes affecting one-carbon metabolism, specifically those affecting the production and use of s-adenosyl methionine (SAM). The depletion of several of these genes has led to cell death by apoptosis in CIN cells but not in normal proliferating cells. We found that CIN cells are particularly sensitive to SAM metabolism at least partly because of its role in generating polyamines. Feeding animals spermine was seen to rescue the cell death caused by the loss of SAM synthase in CIN tissues. The loss of polyamines led to decreased rates of autophagy and sensitivity to reactive oxygen species (ROS), which we have shown to contribute significantly to cell death in CIN cells. These findings suggest that a well-tolerated metabolic intervention such as polyamine inhibition has the potential to target CIN tumours via a relatively well-characterised mechanism.

## 1. Introduction

Chromosomal instability (CIN) refers to karyotype variation that occurs over time, mainly due to a lack of mitotic fidelity [1]. It causes aneuploidy, which is characterised by a deviation from a normal chromosomal number, with either a gain or loss of DNA. CIN is a frequent characteristic of human tumours that causes increasing genetic variation, which has been linked to tumour evolution, medication resistance, and poor prognosis in CIN cancer patients [2]. Aneuploidy is a common hallmark of advanced malignancies, with chromosomal abnormalities detectable in more than 80% of solid tumours [3]. We and others have suggested that because CIN is a cancer-specific trait, it could be a good target for chemotherapy [4,5,6,7]. The process by which CIN is acquired by cancer cells, as well as how it impacts the tolerance of and response to cellular and environmental stressors, are therefore important research goals.

Because CIN cells are genetically varied if not unique, identifying a conserved feature of CIN cells as a prospective target is challenging. However, CIN, and aneuploidy itself, is a common tumour specific phenotype that allows the prospect of tumour specific therapies. To achieve this, it will be necessary to identify the characteristics of aneuploid cells that remain constant regardless of which DNA has been gained or lost. Specifically, we aim to identify what gene expression profiles are essential for the survival of CIN cells, but not normal cells, so that the expression of these genes may become future therapeutic targets.

To find genes that can be depleted to kill CIN cells without affecting normal proliferating cells, we used *Drosophila* to produce CIN in a genetically homogeneous population of cells in vivo [7,8,9]. By weakening the spindle checkpoint using *mad*2-RNAi, we were able to induce CIN. This shortens mitosis and leaves less time for any chromosome misorientation to be corrected during metaphase [10], leading to a considerably higher rate of chromosome segregation errors or CIN [7]. In this flat proliferating epithelium, we typically obtain an average of 25% of mitoses showing an anaphase error without inducing excessive cell death [2]. Using this CIN model system, we have identified several signalling pathways that show promise for therapeutic intervention, particularly metabolic pathways [8,9,11,12].

CIN is sensitive to several processes including proteostasis, redox balance, autophagy, and metabolism [5,8,9,11,13]. CIN induction results in metabolic stress sensitivity [8]. CIN cells are already dealing with high levels of stress, so a small metabolic change that does not harm normal cells can produce large levels of oxidative stress and, ultimately, cell death [8]. For example, metabolic changes that cause decreased NAD+ and more reactive oxygen species (ROS) in aneuploid cells strongly impact their survival [11]. Oxidative stress occurs in response to aneuploidy in all eukaryotes that have been tested, including yeast, plants, *Drosophila*, mice, and humans [11,14]. Previous research has shown an increase in repair mechanisms [9], autophagy [5], and antioxidant levels [8] in response to aneuploidy, which are required to tolerate the deleterious effects of aneuploidy. The induction of aneuploidy increases cells’ metabolic rates, making them vulnerable to oxidative stress so they show DNA damage and apoptosis in response to metabolic challenges that do not damage normal cells [8]. Moreover, aneuploid cells are metabolically different, with overactive mitochondria giving elevated ROS levels [8], which causes damage to macromolecules, resulting in protein-folding defects and ER stress [15]. It has been observed that increased chaperone levels and ER stress markers in aneuploid cells are dependent on ROS levels.

Currently, the mechanisms that regulate the change in metabolism in response to aneuploidy are unclear. Moreover, several aneuploidy-responsive processes, such as autophagy, redox stress, and proteostasis, are too broad to be good therapeutic targets. Therefore, our purpose was to find specific genes/metabolic targets that can be used to kill CIN cells without affecting the normal cells. 

We carried out RNA sequencing to find the genes or metabolic pathways that were altered in normal cells when we induced CIN. We found that the expression of one-carbon and polyamine metabolic genes is significantly upregulated in chromosomal instability (CIN) cells. We aimed to unravel the causal links between aneuploidy and these failures of homoeostasis caused by CIN and its consequences. We investigated the role of autophagy and ROS in mediating one-carbon sensitivity in CIN cells. Moreover, we saw an unexpectedly significant rescue of CIN cells with disruption to their one-carbon metabolism simply by feeding additional polyamines in the diet. 

Overall, our findings suggest that CIN cells are sensitive to one-carbon and polyamine metabolism and that targeting these pathways may represent a promising approach for the development of new strategies for treating and preventing diseases associated with chromosomal instability.

## 2. Materials and Methods

### 2.1. RNA-Seq

Total RNA was extracted from third instar larval tissue using RNEasy Plus minipreps (Qiagen). Six preps using approximately fifteen animals each were generated for each genotype, CIN (*mad*2^EY^/Df(3L)BSC438) vs. non-CIN (*mad*2^EY/+^), and sent for sequencing (150 bp paired end on NovaSeq using Illumina rRNA-depleted TrueSeq libraries). Analysis was carried out using CLC Genomics Workbench 20 (Qiagen). Approximately 40 M sequences per sample were trimmed for adapters, and reads below 15 bp were discarded before mapping to Drosophila genome BDGP6.22.98. The sequences have all been deposited in GEO (accession number GSE231601). The parameters used were default conditions: both strands; no global alignment; max. 10 hits per read; paired reads count as one. An average of 91.4% of reads were mapped in pairs, and 76.2% of reads were in annotated genes. Gene expression changes between CIN and non-CIN samples were quantified in Workbench both for whole genes (GE) and for individual splice variants (TE) and ranked by fold change following an FDR *p* value cutoff of 0.05 (Appendix A). One sample from each genotype was discarded during quality control. The heat map was generated using the average Manhattan distances between clusters, which were filtered using a minimum FDR *p* value of 0.05 and a minimum fold change of 1.5. Gene ontologies of groups significantly enriched or depleted in the CIN genotype were compared using modEnrichr [16]. 

### 2.2. Drosophila Stocks

Stocks were obtained from the Bloomington Stock Center unless otherwise indicated. For the cell death assays in Appendix A, the following stocks were used: *engrailed*-Gal4 (30564), UAS-*mad*2^RNAi^ (VDRC 47918), UAS-*Sam-s*^RNAi^ (29415), UAS-*spds*^RNAi^ (56011), UAS-*sms*^RNAi^ (52924), UAS-*Gnmt*^RNAi^ (53282), UAS-*Ahcy*^RNAi^ (51477), UAS-*Odc*1^RNAi^ (64498), UAS-*PAOX*^RNAi^ (36904), UAS-*Met-s*^RNAi^ (43986), UAS-*Cbs*^RNAi^ (43986), UAS-*Eip*55*E*^RNAi^ (36766), UAS-*SAMdc*^RNAi^ (VDRC 101753), and Df(3L)BSC438 (24942). For the autophagy assays, the following stocks were used: UAS-*Atg*1^RNAi^ (26731), UAS-*mTor*^RNAi^ (33951-scute removed), and UAS-*Atg*18*a*^RNAi^ (34714).

### 2.3. Acridine Orange Staining

The extent of cell death in the engrailed induced third instar larval wing discs was determined using acridine orange (Invitrogen). Wing discs were dissected in PBS from third instar larvae and then stained for 2 min in a 1 μM acridine orange solution, rinsed briefly, mounted, and imaged in PBS. The stain was normalised using ImageJ software by subtracting the average acridine orange signal in the wild type anterior compartment from the average acridine orange signal in the engrailed Gal4-induced mutant posterior compartment (marked with mCD8-GFP). Subtracting the background, a rolling ball radius of 10 pixels or 50 pixels was used [17]. Multiple experiments were carried out, and reproducibility was confirmed by *t*-tests before grouping data.

### 2.4. Immunostaining

In this work, we used our lab’s usual immunostaining procedure [13]. Wing discs were dissected in PBS, fixed in 3.7 percent formaldehyde for 20 min, blocked in PBS plus 0.2 percent Tween-20, and then incubated with primary antibodies overnight at 4 °C. Before and after secondary antibodies were applied, discs were rinsed in PBSTw for 2 h at room temperature. Then, discs were mounted in 80 percent glycerol. All of the images are of larval wing discs in the third instar. The antibodies utilised in this research are listed below: Rabbit anti-DCP-1 (D175, 1:100) is the main antibody (cell signalling). The secondary antibody is rabbit-specific CY3 (1:200). In order to normalise the DCP-1 labelling, the average signal in the wild-type anterior compartment was subtracted from the average signal in the engrailed Gal4-driven mutant posterior compartment (marked with mCD8-GFP). 

### 2.5. Image Data Analysis and Statistics

More information on background subtraction for DCP-1, acridine orange staining, and normalizing the signal from half wing discs to account for variations in staining intensity can be found in [8,12]. GraphPad Prism was utilised for statistical analysis, and ImageJ was used for quantification. The difference in means was calculated using two-tailed t tests with Welsh’s correction, and all error bars indicated 95 percent confidence intervals for the mean. In cases in which multiple comparisons were needed, Dunnett’s T3 method was used, comparing samples to the negative control. 

### 2.6. Drug Treatments

Unless otherwise specified, drugs were purchased from Sigma. Drugs were mixed with common fly food for larvae (water, molasses, yeast, glucose, acid mix, agar, semolina, and Tegosept), and when the mixture solidified, they were administered to the host fly. The medications utilised were N-acetylcysteine amide (NACA) 80 µg/mL, spermine 0.1 mM, and spermidine 0.1 mM and 1 mM. 

### 2.7. ROS Detection and Quantification

As stated, ROS detection was carried out [18]. Briefly, 1 mg of DHE was dissolved in 100 μL DMSO and diluted in Schneider’s medium (SM) before use. The wing discs of dissected larvae were treated with 30 μM DHE in the dark for 15 min after being placed in SM at room temperature. Wing discs were promptly mounted on glass slides with VECTASHIELD Antifade Mounting Medium after being cleaned with PBS four times (Vector Laboratories). Imaging was completed within 4 h.

## 3. Results

### 3.1. The Expression of One-Carbon Metabolism Genes Was Affected by Chromosomal Instability (CIN) and Affected CIN Tolerance 

To find new metabolic pathways altered in response to CIN, we carried out *RNA-Seq* in animals in which CIN had been induced by *mad*2 mutations. A heat map shows consistent changes in gene expression in independent groups of animals of either genotype (Figure 1A). Gene enrichment analysis by modEnrichr shows the top 10 gene ontology hits for biological processes in the GeneRIF library ranked by combined score. Only the top hit, methionine metabolic process (GO:0006555), gave a statistically significant score when corrected for multiple comparisons (Figure 1B). Several genes involved in the methionine or one-carbon cycle showed an elevated transcription in CIN animals (bold in Figure 1C), the strongest hit being S-adenosyl methionine synthase (*SAM-s*). These genes regulate the production of S-adenosyl methionine (SAM), which is critical for several cellular processes, including methylation, antioxidant synthesis, and polyamine synthesis. Our next step was to test whether these genes were significant for the survival of CIN cells. 

We have previously generated models with inducible CIN expression using the RNA interference knockdown of the cohesin gene *Rad*21 or the spindle assembly checkpoint gene *mad*2 [3]. To test whether any of the one-carbon and polyamine regulatory genes affected CIN cell survival, we depleted them by RNAi in a proliferating epithelium (the developing wing disc) in which we induced chromosomal instability (Figure 1D). We observed elevated cell death in these CIN wing discs for several one-carbon regulatory genes, including *SAM-s* (Figure 1D), which is consistent with this pathway having a significant role in CIN cell survival.

### 3.2. Knockdown of Candidates of One-Carbon and Polyamine Pathways Caused CIN Cell Death via Apoptosis

To test whether normal cells are similarly sensitive to one-carbon metabolic disruption, we depleted several candidate genes with and without CIN induction (Figure 2A). We observed a significantly elevated level of cell death in CIN cells compared to adjacent normally proliferating cells (*p* < 0.0001 in each case). This suggests that this pathway can be targeted to give death specifically in CIN cells. However, the assay used (acridine orange) did not clearly indicate the mechanism of cell death (apoptosis vs. necrosis, etc.).

To test whether one-carbon and polyamine metabolic disruption induced apoptosis in CIN cells, we used DCP-1 antibody staining (Figure 2B). We observed considerable induction of apoptosis in proliferating CIN tissues when one-carbon genes were depleted. No induction of apoptosis was observed when CIN was induced alone or when one-carbon genes were depleted in normal proliferating cells. (Figure 2B). The quantification of these data confirmed the induction of CIN-specific apoptosis in candidate gene knockdown cells (Figure 2C). The depletion of some genes in related pathways, such as the folate cycle gene *Shmt*, showed cell death even in the absence of CIN, but this was not observed in the one-carbon or polyamine synthesis pathways. These experiments provided good evidence that one-carbon and polyamine-related candidate gene knockdowns cause CIN-specific cell death via apoptosis.

### 3.3. Feeding of Polyamines Rescued CIN Cell Death Caused by SAM-s or sms Depletion 

The depletion of *SAM-s* or *spds* should decrease the level of spermidine, and the knockdown of *sms* will similarly decrease the synthesis of spermine (see Figure 1C). Because the depletion of any of these enzymes showed death in CIN cells, it seemed plausible that some of the effect of *SAM-s* loss was being mediated by its effect on polyamine synthesis. To test this, we fed spermine to larvae knocked down for *SAM-s* and *sms* in cells with CIN induced by *Mad*2 depletion. We observed that feeding spermine significantly rescued the AO cell death phenotype caused by the depletion of *SAM-s* in CIN cells. (Figure 3A,B). As expected, the feeding of spermine was able to strongly rescue the depletion of *spermine synthase* in CIN cells (Figure 3C,D). A similar rescue of the CIN cell death phenotype caused by *SAM-s* knockdown was seen when larvae were fed with spermidine (Appendix A).

### 3.4. Antioxidant Feeding Rescued CIN Cell Death Phenotypes Caused by One-Carbon or Polyamine Metabolic Gene Depletion

One of the roles of S-Adenosyl methionine is to provide homocysteine for antioxidant synthesis (for glutathione, see Figure 1C). Knowing that redox stress is a significant vulnerability of CIN cells [4], it was plausible that the loss of antioxidant response might be contributing to CIN cells’ vulnerability to *SAM-s* depletion. We carried out ROS assays to examine whether the depletion of *SAM-s* may elevate oxidative stress in CIN cells. We found that the knockdown of *SAM-s* showed an increase in ROS in CIN cells (Figure 4A,B). Having observed elevated cell death in this genotype (Figure 2), we tested whether ROS were contributing to this cell death by feeding the animals an antioxidant (N-acetyl cysteinamide, NACA). We found that NACA feeding significantly reduced the CIN cell death phenotype compared to no drug feeding in the wing disc of third instar larvae (Figure 4C,D). This was consistent with S-adenosyl methionine being a vital antioxidant precursor molecule in CIN cells. However, we had seen a strong rescue of the *Sam-s* phenotype by feeding spermine (Figure 3), so we postulated whether the redox stress might be resulting from the disruption of the polyamine pathway rather than from the lack of homocysteine. To test this, we checked ROS levels in *sms*-depleted CIN cells and found that they were elevated (Figure 4F) compared to in control CIN cells (Figure 4A). We did the same antioxidant feeding experiment with the *sms* mutant animals and found that NACA significantly rescued the cell death seen in *sms*-depleted CIN cells (Figure 4G–I). From these data, we conclude that the depletion of either *SAM-s* or *sms* leads to redox stress in CIN cells, which significantly contributes to the cell death seen in these genotypes.

### 3.5. CIN Cell Death from SAM-s and sms Knockdown Responded to Autophagy

We previously found that autophagy is crucial for CIN cells’ survival and that increasing autophagy could prevent CIN cell death in response to metabolic disruptions [5]. To test whether autophagy also affected *SAM-s*-depleted CIN cells, we checked cell death levels when we increased or decreased autophagy genetically (Figure 5). We found that decreasing autophagy by depleting *Atg*1 caused an increase in the AO staining seen in *SAM-s*-depleted CIN cells (Figure 5C,D). We also found that depleting *SAM-s* with *Atg*18 was synthetically lethal in CIN animals. Although this change is in the direction we would predict, it could have been caused by an accumulation of unrelated cellular disruptions. To demonstrate that the effect of *SAM-s* knockdown depends on autophagy, we increased autophagy by *mTor* RNAi knockdown. Releasing autophagy from *mTOR* inhibition strongly rescued the CIN cell death phenotype caused by *SAM-s* RNAi (Figure 5E vs. Figure 5C).

## 4. Discussion

Aneuploidy resulting from CIN is a common feature of solid tumours [6]. Abnormal cell development, proliferation, proteotoxic stress, and oxidative stress can all be caused by aneuploidy [7,8]. As the adaptability to these various stresses is necessary for survival during continual chromosomal gain or loss, cellular stress responses are a plausible candidate for being the target of cancer-specific apoptosis. The questions then becomes whether CIN cells inevitably approach a tolerance threshold as a result of the constraints of high energy utilisation, ROS formation, proliferation, and continuing genotoxic stress and whether we can leverage this vulnerability.

The present study investigated the hypothesis that alterations in one-carbon and polyamine metabolism can contribute to surviving chromosomal instability (CIN), a hallmark of cancer. This study was based on our RNA sequencing results, which showed an increased transcription of one-carbon and polyamine genes in CIN cells compared to the wild type. Our results provide evidence that supports this hypothesis and suggests that targeting these metabolic pathways could represent a promising strategy for the development of novel cancer therapies. We report that the suppression of one-carbon and polyamine activity in wing imaginal disc cells exhibiting chromosomal instability (CIN) results in highly elevated levels of cell death not observed with either CIN or candidate gene knockdown alone. From these results, we conclude that one-carbon and polyamine metabolism are needed for the survival of CIN cells. Specifically, we found that *SAM-s* or *sms* knockdowns killed CIN cells but not normal proliferating cells and that the severity of these effects depended on oxidative stress and autophagy. We found that polyamine or *SAM-s*-depleted CIN cells showed oxidative stress and that their survival could be significantly enhanced by either adding antioxidants or increasing the level of autophagy. 

The methionine cycle, which is connected to the folate cycle, polyamine synthesis, and the trans-sulphuration route, is a key component of one-carbon metabolism (Figure 1C) [9,10]. S-adenosyl methionine synthase (*SAM-s*) is the enzyme used to create S-adenosylmethionine (SAM) from methionine in the first phase of the methionine cycle. SAM is used for three main purposes: the methylation of protein and DNA substrates, the generation of antioxidants such as cysteine and glutathione, and the synthesis of polyamines. Any of these processes could have been the cause of the cell death seen when SAM was depleted in the CIN cells, so we used a genetic approach to identify what was most important. 

SAM is needed to maintain glutathione and glutathione-S-transferase (GST) levels as well as SOD activity [11]. Hence, it appears likely that one important function of folate and methionine is to enable the efficient regeneration of SAM and, subsequently, glutathione when subjected to oxidative stress. Reactive oxygen species (ROS) levels influence how cancer develops by either supporting carcinogenesis at low levels or by causing cell death at high ones [12]. Due to their abnormal metabolism, tumour cells frequently produce many ROS and have evolved to withstand oxidative stress via a variety of mechanisms, including the production of antioxidants such as glutathione. Even cells that are otherwise normal become redox-stressed when CIN is induced due to a range of mechanisms including ER stress and mitochondrial disruption [2,4,13,14,15,16,17,18]. It seemed plausible that a key reason for needing SAM in CIN cells was to maintain glutathione levels. However, our data cast some doubt on this. We did not see the increased transcription of key enzymes in that pathway (such as cystathionine synthase, *Cbs*) in CIN cells nor were CIN cells sensitive to the depletion of *Cbs* or cystathionine lyase, *Eip55E* (Figure 1D). Nonetheless, we saw the rescue of the *SAM-s* phenotype in CIN cells when antioxidants were provided in the form of NACA feeding, and we show clear evidence for redox stress when SAM is depleted.

The second pathway that depends on SAM is methylation. There are over a hundred methylase enzymes that can use SAM to pass a methyl group onto a huge array of possible substrates [19]. Any of these could mediate the dependency on *SAM-s*; however, we saw a transcriptional response to CIN in only one enzyme, namely, glycine n-methyltransferase, *Gnmt* (Figure 1C), which is also needed for CIN cell survival (Figure 2). The metabolic consequences of *Gnmt* depletion are not obvious as its product, sarcosine, has no known metabolic roles [20]. Current models for the significance of *Gnmt* include its effect on autophagy and polyamine levels [21], which encouraged us to pursue these as possible explanations (see below).

The final metabolic pathway that depends on SAM is polyamine synthesis. SAM is decarboxylated by SAM decarboxylase (*SAMdc*), which produces s-adenosyl methioninamine or dcSAM, a crucial aminopropyl donor for the synthesis of spermidine. The other component of spermidine is produced via putrescine from ornithine via ornithine decarboxylase (ODC). Spermidine synthase produces spermidine by combining putrescine with dcSAM. The last step in this pathway route, spermine, is produced by spermine synthase by combining spermidine with an additional aminopropyl group from dcSAM. We saw a modest increase in *SAMdc* transcription in CIN animals, and a strong, reproducible dependency on polyamine synthesis for CIN survival in multiple assays (Figure 1, Figure 2, Figure 3 and Figure 4). SAM is needed for polyamines (spermine and spermidine), and previous work has shown that these polyamines can affect rates of autophagy [22,23,24]. In flies, yeast, worms, and human cells, spermidine has been shown to trigger autophagy [23,25]. For example, spermidine regulates autophagy by modifying the expression of autophagy-related genes (Atgs) and the gene expression regulators elF5A and TFEB [26,27]. Spermidine also lowers acetyl-CoA availability, which diminishes acetylation and increases autophagy [25]. Polyamines interact with DNA and RNA and are strongly implicated in DNA condensation. This may be relevant as polyamine compaction can protect genomic integrity [28]. Polyamines have other functions, including as antioxidants and for protein synthesis, and CIN cells may also be dependent on these to some degree; however, we saw a strong effect by altering autophagy. Most strikingly, we were able to rescue the depletion of *SAM-s* either by adding spermine to the diet or, even more strongly, by increasing autophagy. These results strongly suggest that the primary dependency on SAM in CIN cells is not for methylation or for glutathione synthesis but rather to generate sufficient polyamines to support robust autophagy.

Our study also highlights the potential therapeutic implications of targeting one-carbon and polyamine metabolism in CIN-associated cancers. SAM has been investigated as a therapeutic target in the treatment of cancer because aberrant methylation states and methionine or SAM dependence are frequent characteristics of cancer [20,29,30,31,32,33,34,35,36]. Additionally, rats given SAM showed an increase in apoptosis in neoplastic nodules and hepatocellular carcinoma [31,34]. SAM treatment increased apoptosis and reduced proliferation and the invasiveness of breast cancer cells in vitro while lowering carcinogenesis and metastasis in vivo [37]. The ability of several cancer cell lines to proliferate and spread has been demonstrated to be effectively inhibited by *SAM* [38,39]. This evidence points to the need for homeostasis: on one hand, many cancers are methionine- and SAM-dependent, but, on the other, they may be inhibited by additional SAM. This may be partly explained by the effect of SAM levels on autophagy as elevated SAM can increase mTORC levels, which blocks autophagy [40]. The loss of SAM means the loss of polyamine synthesis and, hence, the loss of autophagy as described above. Thus, the maintenance of SAM metabolism within normal ranges is important to avoid disruptions to autophagy. 

Some phenotypic effects of CIN are reproducible, even though the genetic disruption is essentially random in each cell [2,41]. There is evidence that chromosomal instability causes sensitivity to metabolic stress [4], protein folding stress, and nucleotide depletion [13]. Proteotoxic stress, faulty mitochondria, and oxidative damage from ROS are three associated abnormalities reliably seen in aneuploid cells that autophagy has been shown to ameliorate [4,42]. To that list, we now add one-carbon and polyamine metabolism. From our results, it seems that cells with chromosomal instability upregulate genes that generate SAM and its polyamine metabolites primarily to increase autophagy, which is needed to survive the stressful metabolic environment of CIN cells.

In conclusion, our study provides new insights into the molecular mechanisms underlying CIN and cancer progression and highlights the importance of one-carbon and polyamine metabolism in regulating autophagy and oxidative stress in CIN cells. Our findings have potential clinical implications for the treatment of CIN-associated cancers, and further studies are warranted to explore the therapeutic potential of targeting one-carbon and polyamine metabolism in combination with other therapies.

## Figures and Tables

**Figure 1 metabolites-13-00642-f001:**
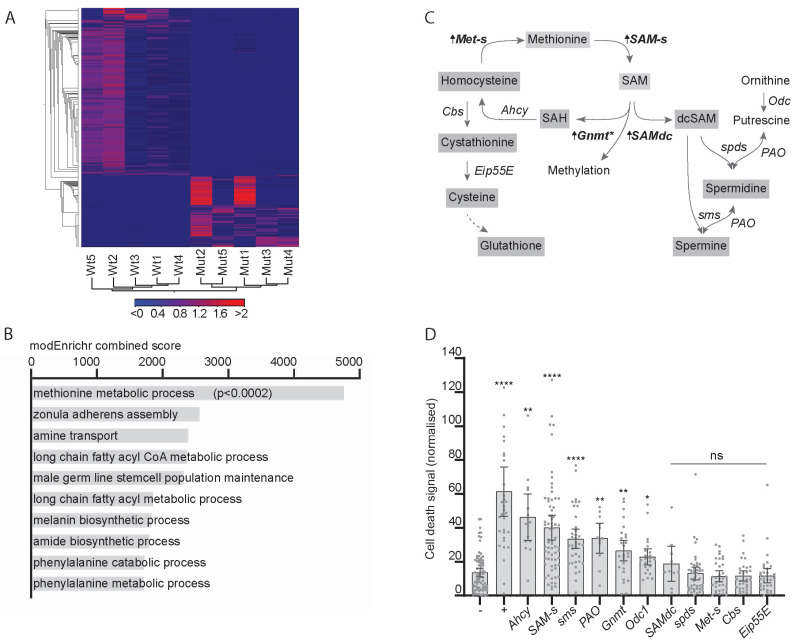
The expression of one-carbon metabolism genes is affected by chromosomal instability (CIN) and affects CIN tolerance. (**A**) Heat map showing changes in gene expression induced by loss of *mad*2. Mutant (*mad*2*^EY^*/Df(2R)BSC438) and wild type (*mad*2*^EY^*/TM6b) samples show distinct sets of transcriptional changes when clustering by genotype. The scale bar shows the shades of grey representing Z-scores for the significance of changes between genotypes in the heat map. (**B**) Gene enrichment analysis by modEnrichr shows the top 10 gene ontology hits for biological processes in the GeneRIF library ranked by combined score. Only the top hit, methionine metabolic process (GO:0006555), gave a statistically significant score when corrected for multiple comparisons. (**C**) One-carbon metabolism pathways, with arrows and bold type indicating changes in transcript expression in response to loss of *Mad*2. *SAM-s*, s-Adenosyl methionine synthase, FDR *p* < 0.0001; *Gnmt*, Glycine n-methyl transferase, FDR *p* < 0.01; SAM, s-Adenosyl methionine; SAH, s-Adenosyl homocysteine; *Ahcy*, Adenosyl homocysteinase; *Met-s*, methionine synthase, FDR *p* < 0.02; Eip55E, Cystathionine lyase; SAMdc, S-Adenosyl methionine decarboxylase, FDR *p* < 0.07; dcSAM, decarboxy-s-Adenosyl methionine. * There are over one hundred *Drosophila* genes with this predicted enzymatic activity (methyl transferases), of which only *Gnmt* expression is significantly elevated in *mad*2 mutants. (**D**) Cell death in third instar wing discs in response to depletion of *mad*2 in combination with one-carbon metabolic gene depletion. Cell death is measured by acridine orange signal normalised to adjacent wild type tissue. Four data points lie outside the graph area. Negative control (−) shows little cell death in response to CIN alone (*w*^1118^ x *en* > *mad*2^RNAi^). Positive control (+) shows significant cell death in CIN cells (JNK is depleted: UAS-*bsk*^RNAi^ x *en* > *mad*2^RNAi^; *p* < 0.001). Depletion of expression of some genes related to one-carbon metabolism showed significant cell death in CIN cells compared to the negative control (**** *p* < 0.001; ** *p* < 0.01; * *p* < 0.05; ns > 0.05 using Dunnett’s T3 multiple comparisons test).

**Figure 2 metabolites-13-00642-f002:**
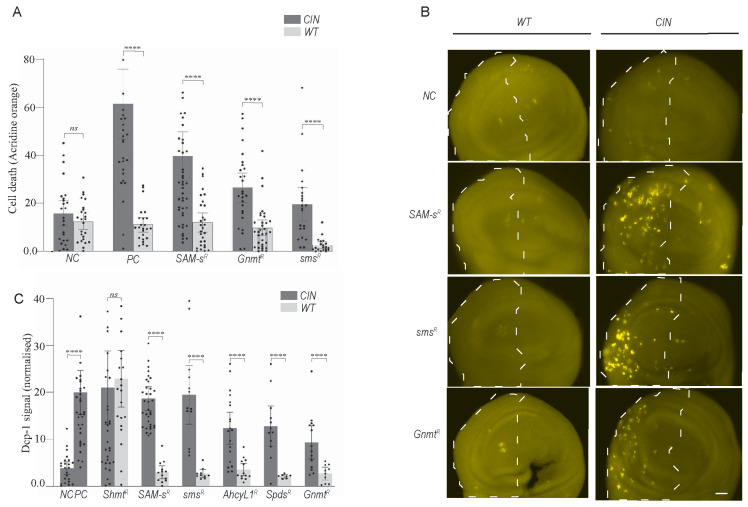
Depletion of one-carbon and polyamine metabolic genes causes CIN cell death but not in normally proliferating cells. (**A**) A quantitative investigation of acridine orange (AO) staining in third instar larvae wing discs with one-carbon metabolic and polyamine pathway genes knocked down with and without CIN. For each wing disc, the y axis depicts the normalised AO signals produced by subtracting the mean value of the control region from the affected region. The error bars show 95% confidence intervals (CIs). (**B**) DCP-1 staining in one-carbon and polyamine candidate RNAi imaginal wing discs with and without CIN. Dashed regions show cells depleted for the candidate gene (and Mad2 in right panels). Original objective magnification, 20×; scale bar = 20 μm. (**C**) The normalised DCP-1 signals for each wing disc are shown on the y axis and are obtained by subtracting the mean value of the control region from the affected region. Shaded bars show one-carbon and polyamine candidate knockdowns in CIN cells, and white bars show the candidate knockdowns in wild type cells. In all graphs, the error bars show 95 percent confidence intervals (CIs). Two-tailed *t*-tests with Welch’s correction were used to generate *p*-values: **** *p <* 0.0001. *NC*: Negative control; *PC*: positive control; *SAM-s*: S-adenosyl methionine synthase; *Gnm*t: Glycine n-methyl transferase; *Sms*: Spermine synthase; *Spds*: spermidine synthase; *Shmt*: Serine hydroxymethyl transferase; *Ahcy*: Adenosyl-homocysteinase.

**Figure 3 metabolites-13-00642-f003:**
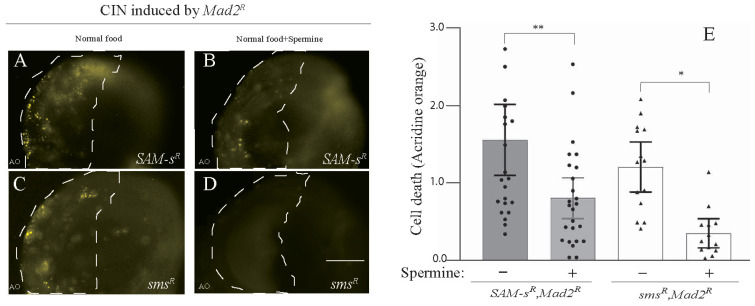
The CIN cell death phenotype caused by the depletion of *SAM-s* and *sms* was rescued by feeding polyamines. The depletion of *SAM-s* (**A**,**B**) and *sms* (**C**,**D**) in CIN cells (induced by *mad*2 RNAi) showed high AO staining, which was rescued by supplementing the larval diet with spermine (0.1 mM). (**E**) *SAM*-S knockdowns in CIN cells are represented by shaded bars with and without spermine, whereas the white bars show *sms* knockdown in CIN cells with and without spermine feeding. In all cases, n ≥ 13, and the error bars represent 95% Cis. The *p*-values were calculated by a multiple comparison test: ** *p* < 0.0053, * *p* < 0.0245. The scale bar is 50 μm, and the original magnification is 20×. Spermine − represents normal food; spermine + represents normal food + 0.1 mM spermine.

**Figure 4 metabolites-13-00642-f004:**
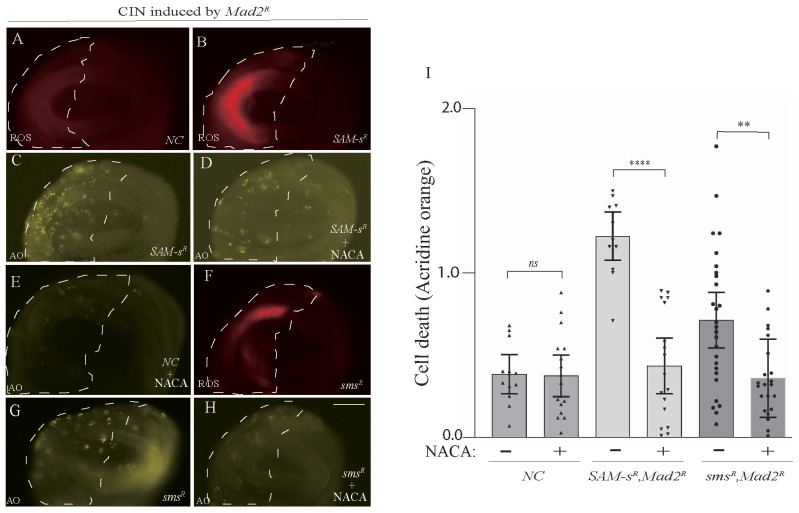
CIN cells were sensitive to polyamine levels because of their effect on the production of reactive oxygen species. ROS levels (DHE staining) were elevated in *Sam-s*-depleted CIN cells (**B**) compared to control CIN cells (**A**). Cell death (acridine orange staining) was decreased by feeding *SAM-s*-depleted CIN cells with the antioxidant N-acetyl Cysteinamide (NACA; compare (**C**) with (**D**) and negative control (**E**)). ROS levels are also elevated in *sms*-depleted CIN cells (**F**). NACA feeding is also able to rescue the cell death seen in *sms*-depleted CIN cells (compare (**G**) and (**H**)). Quantification of CIN cell death with and without NACA feeding in *SAM-s*- and *sms*-depleted cells is shown in (**I**). The indicated genes were knocked down in the posterior half of each wing disc as indicated by the dotted line, while the rest of each disc was wild type. The *p*-values were calculated by multiple comparison tests: **** *p* < 0.0001, ** *p* < 0.0020. The scale bar is 50 μm, ns = not significant, and the original magnification is 20×. The different shading of bars indicates different genotypes *(wild type, SAM-s,* and *sms)* with or without NACA drug feeding. In all cases, *n* > 12, and error bars indicate 95% confidence intervals.

**Figure 5 metabolites-13-00642-f005:**
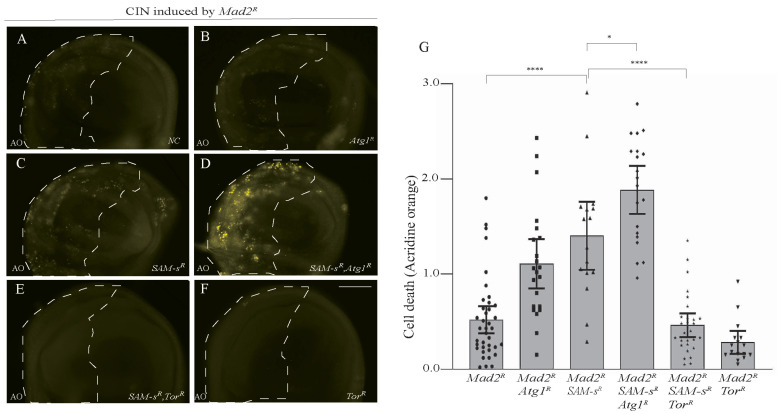
Blocking of autophagy by *Atg1* knockdown increased cell death in *SAM-s*-depleted CIN cells (**A**–**D**). Depletion of *SAM-s* with *Atg1* in CIN cells (induced by *Mad*2 RNAi) showed high AO staining (**D**) compared to only *SAM-s* knockdown (**C**), only *Atg*1 knockdown (**B**), or control CIN cells (**A**). Knockdown of *mTor* in *SAM-s*-depleted CIN cells rescued the cell death phenotype (**E**) to a level resembling the negative control of *mTor* alone (**F**). Quantification and statistical comparison of AO signals is shown in (**G**). In all cases, n ≥ 16, and the error bars represent 95% CIs. The *p*-values were calculated by multiple comparison test: **** *p* < 0.0001, * *p* < 0.0418. The scale bar is 50 μm and the original magnification is 20×.

## Data Availability

The authors may be contacted for more details concerning data supporting the reported results. The data used to support the findings of this study are included within the article.

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
