# Peer review of "Chromosomal Instability Causes Sensitivity to Polyamines and One-Carbon Metabolism"

_metabolites, 2023, doi:10.3390/metabo13050642_

Round 1

Reviewer 1 Report

The manuscript is very interesting. However, I suggest a few points to be improved. I found that the figures in the manuscript are all shown in black-and-white. These figures are very difficult to see and should be replaced with color figures.

The authors used a Drosophila aneuploidy model. They used the model to reconstitute as the CIN (chromosomal instability) phenomenon. Also, they say CIN causes  a lack of mitotic fidelity. The authors should show the relationships between stages and mitotic activity of the tissues used for analyses to convince the aneuploidy indeed causes CIN.   

Author Response

We thank the reviewer for their helpful remarks. It looks like there was a problem with the pdf sent for review - the images were significantly degraded from our originals. We have ensured that the images in the revised manuscript are as they should be, and following the reviewers' suggestions, have coloured them. 

We appreciate that reviewers may not be familiar with the Drosophila models for CIN. Our lab has now published more than ten papers showing the relationship between CIN and aneuploidy in this model, including several reviews. The tissue used is a flat proliferating epithelium in which we induce an anaphase defect in an average of 25% of mitoses (see ref 18). This is a significant amount of CIN - any more than this and we start to see a lot of cell death, and the aneuploid cells are lost. We have modified the text in the introduction to make it clearer that the model we are using is well validated, giving appropriate references.

Reviewer 2 Report

The authors investigated the effect of polyamines on chromosomal instability in Drosophila. They performed RNA-seq in flies in CIN and non-CIN cells, and also investigated separately flies which were fed with spermine as well as those on a normal diet. The authors found that the CIN-associated cells characterized by dysregulation of pathways related to polyamine metabolism. The authors also found that the addition of spermine to the diet decreases apoptosis. Some of the points that need to be improved before publication include the following:

1) The sequencing data reported in this study needs to be released in some open-access database such as GEO.

2) The figures are black and white, but they can be improved with colours.

3) The Methods section can be extended t include details of sequencing data processing, statistics, etc.

4) It would be good to discuss molecular aspects of polyamine interaction with DNA in the process of DNA condensation, which may be involved in the effects studied here.

Author Response

We thank the reviewer for their helpful comments on our manuscript.

1) We have submitted all the RNASeq data to GEO as requested

2) We have replaced the images, they are now coloured as requested

3) We have added more details in the Methods section to make it clear how the sequence data processing was done and the statistical methods used.

4) We have added a section in the discussion relating to the effect of polyamines on DNA condensation and its possible relevance to CIN tolerance.